# Informing policy via dynamic models: Cholera in Haiti

**Jesse Wheeler**[ID][1]*, **AnnaElaine Rosengart**[2], **Zhuoxun Jiang**[1], **Kevin Tan**[3], **Noah Treutle**[1], **Edward L. Ionides**[1]

1 Statistics Department, University of Michigan, Ann Arbor, Michigan, United States of America, 2 Statistics and Data Science, Carnegie Mellon University, Pittsburgh, Pennsylvania, United States of America, 3 Wharton Statistics and Data Science, University of Pennsylvania, Philadelphia, Pennsylvania, United States of America

* jeswheel@umich.edu

**Data Availability Statement:** All data, meta-data and code relevant to the article are publically available at https://doi.org/10.5281/zenodo.10783036 and https://doi.org/10.5281/zenodo.10783080.

## Abstract

Public health decisions must be made about when and how to implement interventions to control an infectious disease epidemic. These decisions should be informed by data on the epidemic as well as current understanding about the transmission dynamics. Such decisions can be posed as statistical questions about scientifically motivated dynamic models. Thus, we encounter the methodological task of building credible, data-informed decisions based on stochastic, partially observed, nonlinear dynamic models. This necessitates addressing the tradeoff between biological fidelity and model simplicity, and the reality of misspecification for models at all levels of complexity. We assess current methodological approaches to these issues via a case study of the 2010-2019 cholera epidemic in Haiti. We consider three dynamic models developed by expert teams to advise on vaccination policies. We evaluate previous methods used for fitting these models, and we demonstrate modified data analysis strategies leading to improved statistical fit. Specifically, we present approaches for diagnosing model misspecification and the consequent development of improved models. Additionally, we demonstrate the utility of recent advances in likelihood maximization for high-dimensional nonlinear dynamic models, enabling likelihood-based inference for spatiotemporal incidence data using this class of models. Our workflow is reproducible and extendable, facilitating future investigations of this disease system.

## Author summary

Quantitative understanding of infectious disease transmission dynamics relies upon mathematical models informed by scientific knowledge and relevant data. The models aim to provide a statistical description of the trajectory of an epidemic and its uncertainty, together with a representation of the underlying biological mechanisms. Evaluation of success at these goals is necessary in order for a model to provide a reliable tool for guiding evidence-based public policy interventions. In this article, we conduct a re-analysis of the 2010–2019 cholera outbreak in Haiti. We use this case study to investigate current procedures for fitting mechanistic models to time series data, while identifying limitations

**Funding:** This work was supported by National Science Foundation (http://www.nsf.gov/div/index.jsp?div=dms) grants DMS-1761603 (EI) and DMS-1646108 (EI). The funders had no role in study design, data collection and analysis, decision to publish, or preparation of the manuscript.

**Competing interests:** The authors have declared that no competing interests exist.

of these methodologies and proposing remedies. Our analysis presents methodology for diagnosing how well a model describes observed data. Using objective measures to assess model fit ensures that our evaluation is based on quantifiable criteria. Incorporating reproducibility into this assessment results in a framework that enables the validation or refinement of model based inferences when revisiting the data, facilitating scientific discovery. Our data analysis workflow is supported by recent advances in algorithms, software and hardware, which facilitate statistical fitting of nonlinear stochastic dynamic models to observed incidence data. However, inference for high-dimensional systems remains a methodological challenge. One of the models under consideration involves spatially coupled stochastic meta-populations, and we demonstrate how a recently developed algorithm permits likelihood-based inference and model diagnostics in this setting. We contend that raising the currently accepted standards of infectious disease modeling will result in a greater ability of scientists and policy makers to understand and respond to future infectious disease outbreaks.

## Introduction

Regulation of biological populations is a fundamental topic in epidemiology, ecology, fisheries and agriculture. Population dynamics may be nonlinear and stochastic, with the resulting complexities compounded by incomplete understanding of the underlying biological mechanisms and by partial observability of the system variables. Quantitative models for these dynamic systems offer potential for designing effective control measures [1, 2]. Developing and testing models for dynamic systems, and assessing their fitness for guiding policy, is a challenging statistical task [3]. Questions of interest include: What indications should we look for in the data to assess whether the model-based inferences are trustworthy? What diagnostic tests and model variations can and should be considered in the course of the data analysis? What are the possible trade-offs of increasing model complexity, such as the inclusion of interactions across spatial units?

This case study investigates the use of dynamic models and spatiotemporal data to inform public health policy in the context of the cholera outbreak in Haiti, which started in 2010. Various dynamic models were developed to study this outbreak: searching PubMed with keywords "Haiti, cholera, model" we obtained 22 studies that utilized deterministic mechanistic dynamic models [4–25] and 11 studies that used stochastic models [4, 26–35]. Incidence data on the outbreak are available at various spatial scales, motivating 17 studies in our literature review to consider spatially explicit dynamic models [4, 5, 8–11, 14, 17, 19, 20, 22–25, 27, 33, 34]. Here we focus on a multi-group modeling exercise by Lee et al. [4] in which four expert modeling teams developed models to the same dataset with the goal of comparing conclusions on the feasibility of eliminating cholera by a vaccination campaign. Model 1 is stochastic and describes cholera at the national level; Model 2 is deterministic with spatial structure, and includes transmission via contaminated water; Model 3 is stochastic with spatial structure, and accounts for measured rainfall. Model 4 has an agent-based construction, featuring considerable mechanistic detail but limited ability to calibrate these details to data. These modeling strategies were selected by Lee et al. [4] to represent the range of approaches used in the research community. We focus on Models 1–3, as the strengths and weaknesses of the agent-based modeling approach [36] are outside the scope of this article. In addition, agent-based models were less widely used, the agent based model in [4] being the only model of this class

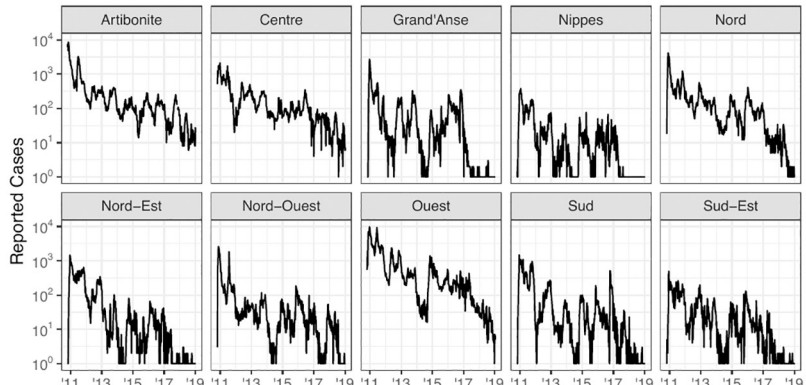

**Fig 1. Weekly cholera cases.** Weekly reported cholera cases in Haiti from October 2010 to January 2019 for each of the 10 administrative departments.

that was found in our literature review. The data that were used in [4], and that we reanalyze, are displayed in Fig 1.

The four independent teams were given the task of estimating the potential effect of prospective oral cholera vaccine (OCV) programs. While OCV is accepted as a safe and effective tool for controlling the spread of cholera, the global stockpile of OCV doses remains limited [37]. Advances in OCV technology and vaccine availability, however, raised the possibility of planning a national vaccination program. The possibility of controlling the Haiti cholera outbreak via OCV was considered by various research groups [4, 6, 11, 12, 22, 35, 38–41]. In the Lee et al. [4] study, certain data were shared between the groups, including demography and vaccination history; vaccine efficacy was also fixed at a shared value between groups. Beyond this, the groups made autonomous decisions on what to include and exclude from their models. Despite their autonomy, the four independent teams obtained a consensus that an extensive nationwide vaccination campaign would be necessary to eliminate cholera from Haiti, estimating that a large number of cumulative cholera cases would be observed in the absence of additional vaccination efforts (Figure 3 and 4 of [4]). These forecasts are inconsistent with the prolonged period with no confirmed cholera cases between February, 2019 and September, 2022 [17]. Though cholera has recently reemerged in Haiti [42, 43], the inability to accurately forecast cholera incidence from 2019–2022 prompts us to consider retrospectively what may have been done differently in order to obtain more reliable conclusions, leading to recommendations for future studies.

The discrepancy between the model-based conclusions of Lee et al. [4] and the prolonged absence of cholera in Haiti has been debated [44–47]. Suggested origins of this discrepancy include the use of unrealistic models [45] and unrealistic criteria for cholera elimination [46]. We find a more nuanced conclusion: attention to methodological details in model fitting, diagnosis and forecasting can improve each of the proposed model's ability to quantitatively describe observed data. This improved ability may result in more accurate forecasts and facilitates the exploration of model assumptions. Based on this retrospective analysis, we offer suggestions on fitting mechanistic models to dynamic systems for future studies.

Numerous guidelines have been proposed for using mechanistic models to inform policy, reviewed in [48]. Behrend et al. [48] identify the importance of stakeholder engagement, transparency, reproducibility, uncertainty communication, and testable model outcomes. These and related principles are echoed by other influential articles [49, 50]. Additional literature

emphasizes model calibration and evaluation techniques [51–53]. These guidelines often lack implementation specifics. As an example, [4] largely adhere to the principles of [48]—though assessing the extent of stake-holder engagement is challenging—yet their projections are inconsistent with actual cholera incidence data from 2019 to 2022, demonstrating the limitations of current standards. We provide methodology for rigorous statistical calibration and evaluation of dynamic models (as advocated by [54]), thereby expanding on the prevailing guidance. We specifically emphasize principles that prove essential in our case study. Complementary methodological suggestions arising from a spatio-temporal analysis of COVID-19 are detailed in [55].

Our recommendations are presented in the context of a case study, with the goal of demonstrating how careful adherence to statistical principles may result in improved model fits. We proceed by introducing the general modeling scheme employed by Models 1–3 and provide details of each individual model; we then describe how each model is calibrated to data, and present a systematic approach to examining and refining these models. Specifically, we focus on how to develop and test variations of the proposed models, as well as diagnosing the models once they have been assimilated to incidence reports. This includes a comprehensive tutorial on performing inference with Model 3 (S5 Text), a highly non-linear, spatially explicit stochastic model, a challenging task that is possible due to recent methodological advancements. We then use the improved model fits to project cholera incidence in Haiti under various vaccination scenarios considered by Lee et al. [4]. Finally, we conclude with a discussion of the results, in which we relate our general recommendations for model based inference of biological systems to the case study of the Haiti cholera outbreak.

## Materials and methods

### Mechanistic models for disease modeling

Mechanistic models representing biological phenomena are valuable for epidemiology and consequently for public health policy [56, 57]. More broadly, they have useful roles throughout biology, especially when combined with statistical methods that properly account for stochasticity and nonlinearity [58]. In some situations, modern machine learning methods can outperform mechanistic models on epidemiological forecasting tasks [59, 60]. The predictive skill of non-mechanistic models can reveal limitations in mechanistic models, but cannot readily replace the scientific understanding obtained by describing the biological dynamics of the system in a mathematical model [60, 61].

In this article, we refer to models that focus on learning relationships between variables in a dataset as *associative*, whereas models that incorporate a known scientific property of the system we call *causal* or *mechanistic*. The danger in using forecasting techniques which rely on associative models to predict the consequence of interventions is called the Lucas critique in an econometric context. Lucas et al. [62] pointed out that it is naive to predict the effects of an intervention on a given system based entirely on historical associations. To successfully predict the effect of an intervention, a model should therefore both provide a quantitative explanation of existing data and should have a causal interpretation: a manipulation of the system should correspond quantitatively with the corresponding change to the model. This motivates the development of mechanistic models, which provides a statistical fit to the available data while also supporting a causal interpretation. Despite their limited ability to project the effect of interventions on a system, associative models can be effectively used to make inference on an certain features of a system. In our literature review, there were 14 studies that used associative models to describe various aspects of the cholera epidemic [9, 40, 63–74].

| Mechanism | Model 1 | Model 2 | Model 3 |
|---|---|---|---|
| Infection (day) | $\mu_{IR}^{-1} = 2.0^\dagger$ (8) | $\mu_{IR}^{-1} = 7.0^\dagger$ (16) | $\mu_{IR}^{-1} = 5.0^\dagger$ (28) |
| Latency (day) | $\mu_{EI}^{-1} = 1.4^\dagger$ (7) | $\mu_{EI}^{-1} = 1.3^\dagger$ (15) | |
| Seasonality | $\beta_{1:6} = (1.4, 1.2, 1.1, 1.1, 1.4, 1.0)$ (4); $\zeta = -0.04^*$ (34) | $a = 0.4^\dagger$ (13); $\phi = 0.97^*$ (13) | $a = 1.00$ (32); $r = 0.78$ (32) |
| Immunity (yr) | $\mu_{RS}^{-1} = 8.0^\dagger$ (9) | $\mu_{RS}^{-1} = 1.4 \times 10^{11}$ (17); $\omega_1^{-1} = 1.0^\dagger$ (19); $\omega_2^{-1} = 5.0^\dagger$ (20) | $\mu_{RS}^{-1} = 8.0^\dagger$ (30) |
| Birth/death (yr⁻¹) | $\mu_S = 10^{-2} \times 2.23^\dagger$ (11); $\delta = 10^{-3} \times 7.5^\dagger$ (11) | | $\delta = 10^{-2} \times 1.59^\dagger$ (29); $\delta_C = 1.46^\dagger$ |
| Sympt. frac. | $f_z(t) = c\vartheta^*(t - \tau_d)^\dagger$ (6-7) | $f = 0.2^\dagger$ (15) | $f = 0.25^\dagger$ (27) |
| Asympt. infectivity | $\epsilon = 0.05^\dagger$ (3) | $\epsilon = 0.001^\dagger$ (13); $\epsilon_W = 10^{-7\dagger}$ (21) | $\epsilon = 1^\dagger$ (25); $\epsilon_W = 0.008$ (32) |
| Human to human | $\beta_{1:6}$ as above (3) | $\beta = 5.97 \times 10^{-15}$ yr⁻¹ (13) | $\beta_{1:10} = (0.82, 0.02, 0.38, 0.21, 0.51, 0.51, 0.35, 0.12, 0.26, 0.10) \times 10^{-6}$ yr⁻¹ (25) |
| Water to human | | $W_{sat} = 10^{5\dagger}$ (13); $\beta_W = 1.1$ yr⁻¹ (13) | $\beta_{W1:10} = (4.70, 21.00, 24.97, 27.14, 5.28, 30.70, 10.17, 0.99, 11.89, 12.82)$ yr⁻¹ (25) |
| Human to water | | $\mu_W = 179$ wk⁻¹ (21) | $\mu_W = 9.77 \times 10^{-7}\ \frac{km^2}{wk}$ (32) |
| Water survival (wk) | | $\delta_W^{-1} = 3^\dagger$ (22) | $\delta_W^{-1} = 0.11$ (33) |
| Mixing exponent | $\nu = 0.98$ (3) | | |
| Process noise (wk^{1/2}) | $\sigma_{proc} = (0.09, 0.12)^*$ (3) | | $\sigma_{proc} = 0.218$ (27) |
| Reporting rate | $\rho = 0.679$ (S3 Text) | $\rho = 0.20^\dagger$ (S3 Text) | $\rho = 0.98$ (S3 Text) |
| Observation variance | $\psi = (279.15, 78.33)$ (S3 Text) | $\psi = 1.319$ (S3 Text) | $\psi = 88.58$ (S3 Text) |
| Initial Values | $I_{0,0} = 7298$; $E_{0,0} = 350$ | | $I_{0,0}^{3,4} = (21, 6)^*$ (S2 Text) |
| Hurricane Parameters | | | $\beta_{W3,9}^{hm} = (36.88, 31.64)^*$ (25); $h_{3,9}^{hm} = (98.98, 58.43)^*$ (25) |

**Fig 2. Model parameters.** References to the relevant equation are given in parentheses. Parameters that were fixed and not calibrated using the data are indicated with $^\dagger$; all fixed parameters values were chosen to match the fixed parameter values of [4]. Parameters that were added during our re-analysis and were not considered by Lee et al. [4] are indicated with $^*$. Confidence intervals for model parameters are given in the supplement (S7 Text). Translations back into the notation of [4] are given in S1 Table.

The four mechanistic models of Lee et al. [4] were deliberately developed with limited coordination. This allows us to treat the models as fairly independently developed expert approaches to understanding cholera transmission. However, it led to differences in notation, and in subsets of the data chosen for analysis, that hinder direct comparison. Here, we have created a common notational framework that facilitates model comparison, and put all comparable model parameters—including parameters that were estimated or held constant—into Fig 2. Translations back to the original notation of Lee et al. [4] are given in the supplement (S1 Table).

Each model describes the cholera dynamics as a partially observed Markov process (POMP) with a latent state vector $X(t)$ for each continuous time point $t$. $N$ observations on the system are collected at time points $t_1, \ldots, t_N$, written as $t_{1:N}$. The observation at time $t_n$ is modeled by the random vector $Y_n$. While the latent process exists between observation times, the value of the latent state at observations times is of particular interest. We therefore write $X_n = X(t_n)$ to denote the value of the latent process at the $n$th observation time, and $X_{1:N}$ is the collection of latent state values for all observed time points. The observable random variables $Y_{1:N}$ are assumed to be conditionally independent given $X_{0:N}$. Together, with the density for the initial value of the latent state $X_0 = X(t_0)$, each model defines a joint density $f_{X_{0:N}, Y_{1:N}}(x_{0:N}, y_{1:N}; \theta)$, where $\theta$ is a parameter vector that indexes the model. The observed data $y_{1:N}^*$, along with the unobserved true value of the latent state, are modeled as a realization of this joint distribution.

Because of the probabilistic nature of both the unobserved latent state and the observable random variables, it is possible to consider various marginal and conditional densities of these two jointly random vectors. An important example is the marginal density of the observed

random vector $Y_{1:N}$, evaluated at the observed data $y_{1:N}^*$, as shown in Eq (1):

$$f_{Y_{1:N}}(y_{1:N}^*; \theta) = \int f_{X_{0:N}, Y_{1:N}}(x_{0:N}, y_{1:N}^*; \theta) dx_{0:N}. \tag{1}$$

When treated as a function of the parameter vector $\theta$, this marginal density is called the *likelihood function*, which is the basis of likelihood based statistical inference.

Using the conditional independence of $Y_{1:N}$ given $X_{0:N}$ and the Markov property of $X_{0:N}$, the joint density can be re-factored into the useful form given in Eq (2):

$$f_{X_{0:N}, Y_{1:N}}(x_{0:N}, y_{1:N}; \theta) = f_{X_0}(x_0; \theta) \prod_{n=1}^{N} f_{X_n | X_{n-1}}(x_n | x_{n-1}; \theta) f_{Y_n | X_n}(y_n | x_n). \tag{2}$$

This factorization is useful because it demonstrates that POMP models may be completely described using three parts: the *initialization model* for the latent states $f_{X_0}(x_0; \theta)$; the *one-step transition density*, or *the process model* $f_{X_n | X_{n-1}}(x_n | x_{n-1}; \theta)$; and the *measurement model* $f_{Y_n | X_n}(y_n | x_n)$. In the following subsections, we describe Models 1–3 in terms of these three components. The latent state vector $X(t)$ for each model consists of individuals labeled as susceptible (S), infected (I), asymptomatically infected (A), vaccinated (V), and recovered (R), with various sub-divisions sometimes considered. The observable random vector $Y_{1:N}$ represents the random vector of cholera incidence data for each model; Models 2 and 3 have metapopulation structure, meaning that each individual is a member of a spatial unit, denoted by a subscript $u \in 1:U$, in which case we denote the observed data for each unit using $y_{1:N}^* = y_{1:N,1:U}^*$. Here, the spatial units are the $U = 10$ Haitian administrative départements (henceforth anglicized as departments).

While the complete model description is scientifically critical, as well as necessary for transparency and reproducibility, the model details are not essential to our methodological discussions of how to diagnose and address model misspecification with the purpose of informing policy. A first-time reader may choose to skim through the rest of this section, and return later. Additional details about the numeric implementation of these models are provided in a supplemental text (S1 Text). While each of the dynamic models considered in this manuscript can be fully described using the mathematical equations provided in the following section, diagrams of dynamic systems can be helpful to understand the equations. For this reason, we provide flow chart diagrams for Models 1–3 in supplement figures (S1–S3 Figs).

**Model 1.** The latent state vector $X(t) = (S_z(t), E_z(t), I_z(t), A_z(t), R_z(t), z \in 0 : Z)$ describes susceptible, latent (exposed), infected (and symptomatic), asymptomatic, and recovered individuals in vaccine cohort $z$ at time $t$. Here, $z = 0$ corresponds to unvaccinated individuals, and $z \in 1:Z$ describes hypothetical vaccination programs. Each program $z$ indexes differences in both the number of doses administered (one versus two doses per individual) and the round of vaccine administration, separating individuals into compartments with distinct dynamics based on vaccination status. The force of infection is

$$\lambda(t) = \left( \sum_{z=0}^{Z} I_z(t) + \epsilon \sum_{z=0}^{Z} A_z(t) \right)^{v} \frac{d\Gamma(t)}{dt} \beta(t) / N, \tag{3}$$

where $\beta(t)$ is a periodic cubic spline representation of seasonality, given in terms of a B-spline basis $\{s_j(t), j \in 1:6\}$ and parameters $\beta_{1:6}$ as

$$\beta(t) = \bar{\beta} \exp(\sum_{j=1}^{6} \beta_j s_j(t)), \tag{4}$$

where $\bar{\beta} = 1$ (wk)$^{-1}$ is a dimensionality constant. The process noise $d\Gamma(t)/dt$ is multiplicative Gamma-distributed white noise, with infinitesimal variance parameter $\sigma_{\text{proc}}^2$. Lee et al. [4] included process noise in Model 3 but not in Model 1, i.e., they fixed $\sigma_{\text{proc}}^2 = 0$. Gamma white noise in the transmission rate gives rise to an over-dispersed latent Markov process [75] which has been found to improve the statistical fit of disease transmission models [2, 76].

For any time point in $t_{1:N}$, the process model $f_{X_n|X_{n-1}}(\boldsymbol{x}_n|\boldsymbol{x}_{n-1}; \theta)$ is defined by describing how individuals move from one latent state compartment to another. Per-capita transition rates are given in Eqs (5)–(12):

$$\mu_{S_z E_z} = \lambda(t), \tag{5}$$

$$\mu_{E_z I_z} = \mu_{EI}(1 - f_z(t)), \tag{6}$$

$$\mu_{E_z A_z} = \mu_{EI} f_z(t), \tag{7}$$

$$\mu_{I_z R_z} = \mu_{A_z R_z} = \mu_{IR}, \tag{8}$$

$$\mu_{R_z S_z} = \mu_{RS}, \tag{9}$$

$$\mu_{S_0 S_z} = \mu_{E_0 E_z} = \mu_{I_0 I_z} = \mu_{A_0 A_z} = \mu_{R_0 R_z} = \eta_z(t), \tag{10}$$

$$\mu_{S_z \bullet} = \mu_{E_z \bullet} = \mu_{I_z \bullet} = \mu_{A_z \bullet} = \mu_{R_z \bullet} = \delta, \tag{11}$$

$$\mu_{\bullet S_0} = \mu_S, \tag{12}$$

where $z \in 0{:}Z$. Here, $\mu_{AB}$ is a transition rate from compartment $A$ to $B$. We have an additional demographic source and sink compartment $\bullet$ modeling entry into the study population due to birth or immigration, and exit from the study population due to death or immigration. Thus, $\mu_{A\bullet}$ is a rate of exiting the study population from compartment $A$ and $\mu_{\bullet B}$ is a rate of entering the study population into compartment $B$.

In Model 1, the advantage afforded to vaccinated individuals is an increased probability that an infection is asymptomatic. Conditional on infection status, vaccinated individuals are also less infectious than their non-vaccinated counterparts by a rate of $\epsilon = 0.05$ in Eq (3). In Eqs (7) and (6) the asymptomatic ratio for non-vaccinated individuals is set $f_0(t) = 0$, so that the asymptomatic route is reserved for vaccinated individuals. For $z \in 1{:}Z$, the vaccination cohort $z$ is assigned a time $\tau_z$, and we take $f_z(t) = c\vartheta^*(t - \tau_z)$ where $\vartheta^*(t)$ is efficacy at time $t$ since vaccination for adults, a step-function represented in Table S4 of [4], and $c = (1 - (1 - 0.4688) \times 0.11)$ is a correction to allow for reduced efficacy in the 11% of the population aged under 5 years. Single and double vaccine doses were modeled by changing the waning of protection; protection was modeled as equal between single and double dose until 52 weeks after vaccination, at which point the single dose becomes ineffective.

The latent state vector $\boldsymbol{X}(t)$ is initialized by setting the counts for each compartment and vaccination scenario $z \neq 0$ as zero, and introducing initial-value parameters $I_{0,0}$ and $E_{0,0}$ such that $R_0(0) = 0$, $I_0(0) = \text{Pop} \times I_{0,0}$, $E_0(0) = \text{Pop} \times E_{0,0}$ and $S_0(0) = \text{Pop} \times (1 - I_{0,0} - E_{0,0})$, where Pop is the total population of Haiti. The measurement model describes reported cholera cases at time point $n$ come from a negative binomial distribution, where only a fraction ($\rho$) of new weekly cases are reported. More details about the initialization model $f_{X_0}(\boldsymbol{x}_0; \theta)$ and the

measurement model $f_{Y_n|X_n}(\boldsymbol{y}_n|\boldsymbol{x}_n)$ for Models 1–3 are provided in a supplement text (S2 and S3 Text).

**Model 2.** Susceptible individuals are in compartments $S_{uz}(t)$, where $u \in 1{:}U$ corresponds to the $U = 10$ departments, and $z \in 0{:}4$ describes vaccination status:

$z = 0$: Unvaccinated or waned vaccination protection.

$z = 1$: One dose at age under five years.

$z = 2$: Two doses at age under five years.

$z = 3$: One dose at age over five years.

$z = 4$: Two doses at age over five years.

Like Model 1, the process model $f_{X_n|X_{n-1}}(\boldsymbol{x}_n|\boldsymbol{x}_{n-1}; \theta)$ is primarily defined via the description of movement of individuals between compartments, however Model 2 also includes a dynamic description of a latent bacterial compartment as well. Individuals can progress to a latent infection $E_{uz}$ followed by symptomatic infection $I_{uz}$ with recovery to $R_{uz}$ or asymptomatic infection $A_{uz}$ with recovery to $R_{uz}^A$. The force of infection depends on both direct transmission and an aquatic reservoir, $W_u(t)$, and is given by

$$\lambda_u(t) = 0.5(1 + a\cos(2\pi t + \phi))\frac{\beta_W W_u(t)}{W_{\text{sat}} + W_u(t)} + \beta\left\{\sum_{z=0}^{4} I_{uz}(t) + \epsilon \sum_{z=0}^{4} A_{uz}(t)\right\}. \tag{13}$$

The latent state is therefore described by the vector $\boldsymbol{X}(t) = (S_{uz}(t), E_{uz}(t), I_{uz}(t), A_{uz}(t), R_{uz}(t), R_{uz}^A(t), W_u, u \in 1{:}U, z \in 0{:}4)$. The cosine term in Eq (13) accounts for annual seasonality, with a phase parameter $\phi$. The Lee et al. [4] implementation of Model 2 fixes $\phi = 0$.

Individuals move from department $u$ to $v$ at rate $T_{uv}$, and aquatic cholera moves at rate $T_{uv}^W$. The nonzero transition rates are

$$\mu_{S_{uz}E_{uz}} = (1 - \vartheta_z)\lambda_u(t), \tag{14}$$

$$\mu_{E_{uz}I_{uz}} = f\mu_{EI}, \quad \mu_{E_{uz}A_{uz}} = (1-f)\mu_{EI}, \tag{15}$$

$$\mu_{I_{uz}R_{uz}} = \mu_{A_{uz}R_{uz}^A} = \mu_{IR}, \tag{16}$$

$$\mu_{R_{uz}S_{uz}} = \mu_{R_{uz}^A S_{uz}} = \mu_{RS}, \tag{17}$$

$$\mu_{S_{uz}S_{vz}} = \mu_{E_{uz}E_{vz}} = \mu_{I_{uz}I_{vz}} = \mu_{A_{uz}A_{vz}} = \mu_{R_{uz}R_{vz}} = \mu_{R_{uz}^A R_{vz}^A} = T_{uv}, \tag{18}$$

$$\mu_{S_{u1}S_{u0}} = \mu_{S_{u3}S_{u0}} = \omega_1, \tag{19}$$

$$\mu_{S_{u2}S_{u0}} = \mu_{S_{u4}S_{u0}} = \omega_2, \tag{20}$$

$$\mu_{\bullet W_u} = \mu_W \left\{ \sum_{z=0}^{4} I_{uz}(t) + \epsilon_W \sum_{z=0}^{4} A_{uz}(t) \right\}, \tag{21}$$

$$\mu_{W_u \bullet} = \delta_W, \tag{22}$$

$$\mu_{W_u W_v} = w_r T_{uv}^W. \tag{23}$$

In Eq (18) the spatial coupling is specified by a gravity model,

$$T_{uv} = v_{\text{rate}} \times \frac{\text{Pop}_u \text{Pop}_v}{D_{uv}^2}, \tag{24}$$

where $\text{Pop}_u$ is the mean population for department $u$, $D_{uv}$ is a distance measure estimating average road distance between randomly chosen members of each population, and $v_{\text{rate}} = 10^{-12} \, \text{km}^2 \text{yr}^{-1}$ was fixed at the value used in [4]. In Eq (23), $T_{uv}^W$ is a measure of river flow between departments. The unit of $W_u(t)$ is cells per ml, with dose response modeled via a saturation constant of $W_{\text{sat}}$ in Eq (13). In Eq (14), $\vartheta_z$ denotes the vaccine efficacy for each vaccination campaign $z \in Z$, with $\vartheta_0 = 0$, $\vartheta_1 = 0.429q$, $\vartheta_2 = 0.519q$, $\vartheta_3 = 0.429$, and $\vartheta_4 = 0.519$ Here, $q = 0.4688$ represents the reduced efficacy of the vaccination for children under the age of five years, and the values 0.429 and 0.519 are the median effectiveness of one and two doses over their effective period respectively, according to Table S4 in the supplement material of Lee et al. [4]. Because vaccine efficacy remains constant, individuals in this model transition from a vaccinated compartment to the susceptible compartment at the end of the vaccine coverage period.

The starting value for each element of the latent state vector $X(0)$ are set to zero except for $I_{u0}(0) = y_u^*(0)/\rho$ and $R_{u0}(0) = \text{Pop}_u - I_{u0}(0)$, where $y_u^*(0)$ is the reported number of cholera cases in department $u$ at time $t = 0$. Reported cases are described using a log-normal distribution, with the log-scale mean equal to the reporting rate $\rho$ times the number of newly infected individuals. See the supplement material on model initializations for more details (S2 Text).

**Model 3.** The latent state is described as $X(t) = (S_{uz}(t), I_u(t), A_u(t), R_{uzk}(t), W_u(t), u \in 0{:}U, z \in 0{:}4, k \in 1{:}3)$. Here, $z = 0$ corresponds to unvaccinated, $z = 2j - 1$ corresponds to a single dose on the $j$th vaccination campaign in unit $u$ and $z = 2j$ corresponds to receiving two doses on the $j$th vaccination campaign. $k \in 1{:}3$ models non-exponential duration in the recovered class before waning of immunity. The processes model $f_{X_n|X_{n-1}}(x_n|x_{n-1}; \theta)$ describes the movement of individuals between latent compartments, as well as the birth and death process of local, unobserved bacterial compartments $W_u(t)$. The force of infection is

$$\lambda_u(t) = \left( \beta_{W_u} + 1_{(t \geq t_{hm})} \beta_{W_u}^{hm} e^{-h_u^{hm}(t - t_{hm})} \right) \frac{W_u(t)}{1 + W_u(t)} + \beta_u \sum_{v \neq u} (I_v(t) + \epsilon A_v(t)), \tag{25}$$

where $t_{hm}$ is the time Hurricane Matthew struck Haiti [77], and $1_{(A)}$ is the indicator function for event $A$. In [4], $\beta_{W_u}^{hm}$ and $h_u^{hm}$ were set to zero for all $u$; the need to account for the effect Hurricane Matthew had on cholera transmission for this model is explored in the supplement (S5 Text).

Per-capita transition rates are used for both compartments representing human counts and the aquatic reservoir of bacteria; these rates are given in Eqs (26)–(33).

$$\mu_{S_{uz}I_u} = f\,\lambda_u(1 - \vartheta_{uz}(t))\,d\Gamma/dt, \tag{26}$$

$$\mu_{S_{uz}A_u} = (1-f)\,\lambda_u(1 - \vartheta_{uz}(t))\,d\Gamma/dt, \tag{27}$$

$$\mu_{I_u R_{uz1}} = \mu_{A_u R_{uz1}} = \mu_{IR}, \tag{28}$$

$$\mu_{I_u S_{u0}} = \delta + \delta_C, \quad \mu_{A_u S_{u0}} = \delta \tag{29}$$

$$\mu_{R_{uz1}R_{uz2}} = \mu_{R_{uz2}R_{uz3}} = 3\mu_{RS}, \tag{30}$$

$$\mu_{R_{uzk}S_{u0}} = \delta + 3\mu_{RS}\,\mathbf{1}_{\{k=3\}}, \tag{31}$$

$$\mu_{\bullet W_u} = [1 + a(J_u(t))^r]\mathrm{Den}_u\,\mu_W[I_u(t) + \epsilon_W A_u(t)], \tag{32}$$

$$\mu_{W_u\bullet} = \delta_W. \tag{33}$$

As with Model 1, $d\Gamma_u(t)/dt$ is multiplicative Gamma-distributed white noise in Eqs (26) and (27). In Eq (32), $J_u(t)$ is a dimensionless measurement of precipitation that has been standardized by dividing the observed rainfall at time $t$ by the maximum recorded rainfall in department $u$ during the epidemic, and $\mathrm{Den}_u$ is the population density. Demographic stochasticity is accounted for by modeling non-cholera related death rate $\delta$ in each compartment, along with an additional death rate $\delta_C$ in Eq (29) to account for cholera induced deaths among infected individuals. All deaths are balanced by births into the susceptible compartment in Eqs (29) and (31), thereby maintaining constant population in each department.

Similar to Model 1, there are no distinct compartments for individuals under five years of age, and the vaccination efficacy is taken as a age adjusted weighted average of the efficacy for individuals both over and under five years of age: $\vartheta_{uz}(t) = c\vartheta^*(t - \tau_{uz})$, where $\tau_{uz}$ is the vaccination time for unit $u$ and vaccination campaign $z$. The value $c$ and the function $\vartheta^*$ are equivalent to those described in the Model 1 description.

Latent states are initialized using an approximation of the instantaneous number of infected, asymptomatic, and recovered individuals at time $t_0$ by using the first week of cholera incidence data. Specifically, we set $I_{u0}(0) = \frac{y^*_{1u}}{\rho(\delta + \delta_C + \mu_{IR})}$, $A_{u0}(0) = \frac{1-f}{f} I_{u0}(0)$, $R_{u0k} = y^*_{1u} - I_{u0}(0) - A_{u0}(0)$, and we initialize $W_u(0)$ by enforcing the rainfall dynamics supposed by the one step transition model; all other compartments that represent population counts are set to zero at time $t_0$. For each unit $u$ with zero case counts at time $t_1$, this initialization scheme results in having zero individuals in the Infected and Asymptomatic compartments, as well as having no bacteria in the aquatic reservoir. In reality, it is plausible that some bacteria or infected individuals were present in unit $u$ but went unreported. Therefore, for departments with zero case counts in week 1, we estimate the number of infected individuals rather then treating this value as a constant (S2 Text). Finally, reported cholera cases are modeled using a negative binomial distribution with mean equal to a fraction ($\rho$) of individuals in each unit who develop symptoms and seek healthcare, and with over-dispersion parameter $\psi$ (S3 Text).

## Model fitting

Each of the three models considered in this study describes cholera dynamics as a partially observed Markov process (POMP) [3], with the understanding that the deterministic Model 2 is a special case of a Markov processes solving a stochastic differential equation in the limit as the noise parameter goes to zero. Each model is indexed by a parameter vector, $\theta$, and different values of $\theta$ can result in qualitative differences in the predicted behavior of the system. Therefore, the choice of $\theta$ used to make inference about the system can greatly affect model-based conclusions [49]. Elements of $\theta$ can be fixed at a constant value based on scientific understanding of the system, but parameters can also be calibrated to data by maximizing a measure of congruency between the observed data and the model's mechanistic structure. Calibrating model parameters to observed data does not guarantee that the resulting model successfully approximates real-world mechanisms, since the model description of the dynamic system may be incorrect and does not change as the model is calibrated to data. However, the congruency between the model and observed data serves as a proxy for the congruency between the model and the true underlying dynamic system. As such, it is desirable to obtain the best possible fit of the proposed mechanistic structure to the observed data.

In this article we follow [4] by calibrating the parameters of each of our models using maximum likelihood, as described in Eq (1). The likelihood for each of the fitted models—and the corresponding AIC values for model comparisons that include an adjustment for the number of calibrated parameters—is provided in Table 1. In the following subsections we describe in detail our approach to calibrating the three proposed mechanistic models to observed cholera incidence data. The main alternative to maximum likelihood estimation is Bayesian inference via Markov chain Monte Carlo, used to analyze the Haiti cholera epidemic by [6, 10, 17, 23–25, 27, 30, 33, 34].

**Calibrating Model 1 parameters.** Model 1 is a highly nonlinear over-dispersed stochastic dynamic model, favoring a scientifically plausible description of cholera dynamics rather than one that is statistically convenient [2]. This results in the inability to obtain a closed form expression of the joint model density—described in Eq (2). Therefore in order to perform likelihood based inference on this model, we are restricted to use parameter estimation techniques that have the *plug-and-play* property, which is that the fitting procedure only requires the ability to simulate the latent process rather than evaluating transition densities [2, 78]; in the context of the notation and definitions employed in this article, this means that we only require

**Table 1. AIC values for Models 1–3 and their benchmarks.**

| | Model 1 | Model 2 | Model 3 |
|---|---|---|---|
| Log-likelihood | −2728.1 | −21957.3 | −17332.9 |
| | (−3030.9)[1] | (−29367.4) | (−33832.6)[2] |
| Number of Fit Parameters | 15 | 6 | 34 |
| | (20) | (6) | (29) |
| AIC | 5486.3 | 43926.5 | 34733.9 |
| | (6101.8)[1] | (58746.9) | (67723.2)[2] |
| Benchmark AIC | 5585.3 | 36961.0 | 35945.2 |

Values in parentheses are corresponding values obtained using the models of [4].

[1]The reported likelihood is an upper bound of the likelihood of the model in [4].

[2]In [4], Model 3 was fit to a subset of the data (March 2014 onward, excluding data from Ouest in 2015–2016). On this subset, their model has a likelihood of −9721.2. On this same subset, our model has a likelihood of −7219.5. Details of estimating the likelihood of the models used in [4] are provided in the supplement (S4 Text).

 

the ability to simulate from $f_{X_0}(\boldsymbol{x}_0; \theta)$ and $f_{X_n|X_{n-1}}(\boldsymbol{x}_n|\boldsymbol{x}_{n-1}; \theta)$ rather than needing to evaluate these densities. Plug-and-play algorithms include Bayesian approaches like ABC and PMCMC [79, 80], but here we use algorithms that enable maximum likelihood estimation. To our knowledge, the only plug-and-play methods that have been effectively used to maximize the likelihood for arbitrary nonlinear POMP models are iterated filtering algorithms [81], which modify the well-known *particle filter* [82].

The particle filter, also referred to as sequential Monte Carlo, is a simulation based method that is frequently used in Bayesian inference to approximate the posterior distribution of latent states. This algorithm can also be used to accurately approximate the log-likelihood of a POMP model, defined as the integral in Eq (1). Iterated filtering algorithms, such as IF2 [81], extend the particle filter by performing a random walk for each parameter and particle; these perturbations are carried out iteratively over multiple filtering operations, using the collection of parameters from the previous filtering pass as the parameter initialization for the next iteration, and decreasing the random walk variance at each step. With a sufficient number of iterations, the resulting parameter values converge to a region of the parameter space that maximizes the model likelihood.

The ability to maximize the likelihood allows for likelihood-based inference, such as performing statistical tests for potential model improvements. We demonstrate this capability by proposing a log-linear trend $\zeta$ in transmission in Eq (4):

$$\beta(t) = \bar{\beta} \exp\left(\sum_{j=1}^{6} \beta_s s_j(t) + \zeta \bar{t}\right), \tag{34}$$

where $\bar{t} = \frac{t - (t_N + t_0)/2}{t_N - (t_N + t_0)/2}$, so that $\bar{t} \in [-1, 1]$. The proposal of a trend in transmission is a result of observing an apparent decrease in reported cholera infections from 2012–2019 in Fig 1. While several factors may contribute to this decrease, one explanation is that case-area targeted interventions (CATIs), which included education sessions, increased monitoring, household decontamination, soap distribution, and water chlorination in infected areas [66], may have substantially reduced cholera transmission over time [83].

We perform a statistical test to determine whether or not the data indicate the presence of a trend in transmissibility. To do this, we perform a profile-likelihood search on the parameter $\zeta$ and obtain a 95% confidence interval via a Monte Carlo Adjusted Profile (MCAP) [84]. Lee et al. [4] implemented Model 1 by fitting two distinct phases: an epidemic phase from October 2010 through March 2015, and an endemic phase from March 2015 onward. We similarly allow the re-estimation of process and measurement overdispersion parameters ($\sigma_{\text{proc}}^2$ and $\psi$), and require that the latent Markov process $X(t)$ carry over from one phase into the next. The resulting 95% confidence interval for $\zeta$ is $(-0.098, -0.009)$, with the full results displayed in Fig 3. These results are suggestive that the inclusion of a trend in the transmission rate improves the quantitative ability of Model 1 to describe the observed data. The maximum likelihood estimate for $\zeta$ corresponds to a 7.3% reduction to the transmission rate over the course of the outbreak, with a 95% confidence interval of (1.8%, 17.9%) for the overall reduction in transmission. The reported results for Model 1 in the remainder of this article were obtained with the inclusion of the parameter $\zeta$. The inclusion of a trend in transmission rate demonstrates a class of model variation that can be highly beneficial to consider: the model variation has a plausible scientific justification, and is easily testable using likelihood based methods.

If a mechanistic model including a feature (such as a representation of a mechanism, or the inclusion of a covariate) fits better than mechanistic models without that feature, and also has competitive fit compared to associative benchmarks, this may be taken as evidence supporting

 

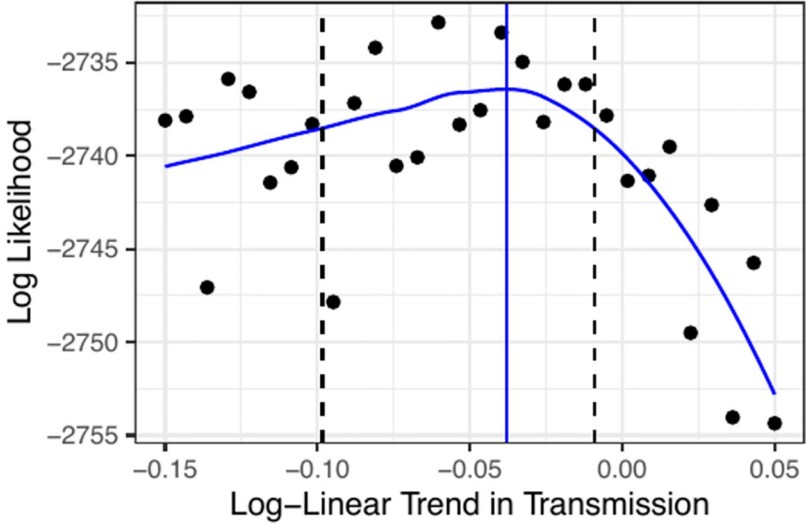

**Fig 3. Confidence interval for the log-linear trend in transmission.** Monte Carlo adjusted profile (MCAP) of $\zeta$ for Model 1. The blue curve is the MCAP, the vertical blue line indicates the MLE, and the vertical dashed lines indicate the 95% confidence interval.

the scientific relevance of the feature. As for any analysis of observational data, we must be alert to the possibility of confounding. For a covariate, this shows up in a similar way to regression analysis: the covariate under investigation could be a proxy for some other unmodeled phenomenon or unmeasured covariate.

The statistical evidence of a trend in transmission rate in this model could be explained by any trending variable (such as hygiene improvements, or changes in population behavior), resulting in confounding from collinear covariates. Alternatively, it is possible that the negative trend observed in the incidence data could be attributed to a decreasing reporting rate rather than decreasing transmission rate. This could be formally tested by comparing models with either trend specification. We did not do this because evidence suggests that reporting rate was maintained or increased (Figure 1 of [83]). We instead argue that a decreasing transmission rate is a plausible way to explain the decrease in cases over time, as there is alternative evidence that supports this model [66, 83, 85]. It is not practical to test all remotely plausible model variations, yet a strongly supported conclusion should avoid ruling out untested hypotheses. The robust statistical conclusion for our analysis is that a model which allows for change fits better than one which does not, and a trend in transmission is a plausible way to do this.

We implemented Model 1 using the `pomp` package [3], relying heavily on the source code provided by Lee et al. [4]. Both analyses used the `mif2` implementation of the IF2 algorithm to estimate $\theta$ by maximum likelihood. One change we made in the statistical analysis that led to larger model likelihoods was increasing the computational effort in the numerical maximization. While IF2 enables parameter estimation for a large class of models, the theoretic ability to maximize the likelihood depends on asymptotics in both the number of particles and the number of filtering iterations. Many Monte Carlo replications are then required to quantify and further reduce the error. The large increase in the log-likelihood for Model 1 (Table 1) can primarily be attributed to increasing the computational effort used to calibrate the model. This result highlights the importance of carefully determining the necessary computational effort needed to maximize model likelihoods and acting accordingly. In this case study, this was done by performing standard diagnostics for the IF2 and particle filter algorithms [3, 55, 86,

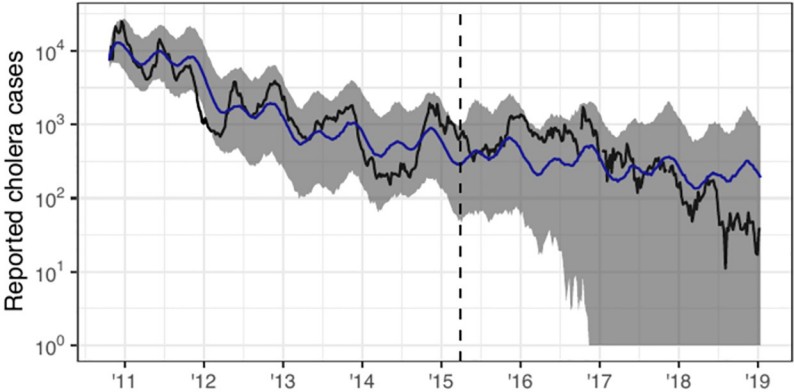

**Fig 4. Simulations from Model 1 compared to reported cholera cases.** The black curve is observed data, the blue curve is median of 500 simulations from initial conditions using estimated parameters, and the vertical dashed line represents break-point when parameters are refit.

87]. Given the considerable computational costs of simulation-based algorithms, we find it useful to perform an initial assessment using hyperparameter values—such as the number of particles, filtering iterations, and replicates based on different parameter initializations—that enable relatively quick calculations. The insights obtained from this preliminary analysis help in accurately determining the amount of computation that is required to achieve reliable outcomes. Simulations from the initial conditions of our fitted model are plotted against the observed incidence data in Fig 4.

**Calibrating Model 2 parameters.** Model 2 is a deterministic compartmental model defined by a set of coupled differential equations. The use of deterministic compartment models have a long history in the field of infectious disease epidemiology [88–90], and can be justified by asymptotic considerations in a large-population limit [91, 92]. Because the process model of Model 2 is deterministic, maximum likelihood estimation reduces to a least squares calculation when combined with a Gaussian measurement model (S3 Text). Lee et al. [4] fit two versions of Model 2 based on a presupposed change in cholera transmission from a epidemic phase to endemic phase that occurred in March, 2014. The inclusion of a change-point in model states and parameters increased the flexibility of the model and hence the ability to fit the observed data. The increase in model flexibility, however, resulted in hidden states that were inconsistent between model phases. The inclusion of a model break-point by Lee et al. [4] is perhaps due to a challenging feature of fitting a deterministic model via least squares: discrepancies between model trajectories and observed case counts in highly infectious periods of a disease outbreak will result in greater penalty than the discrepancies between model trajectories and observed case counts in times of relatively low infectiousness. This results in a bias towards accurately describing periods of high infectiousness. This bias is particularly troublesome for modeling cholera dynamics in Haiti: the inability to accurately fit times of low infectiousness may result in poor model forecasts, as few cases of cholera were observed in the last few years of the epidemic.

To combat this issue, we fit the model to log-transformed case counts, since the log scale stabilizes the variation during periods of high and low incidence. An alternative solution is to change the measurement model to include overdispersion, as was done in Models 1 and 3. This permits the consideration of demographic stochasticity, which is dominant for small infected populations, together with log scale stochasticity (also called multiplicative, or

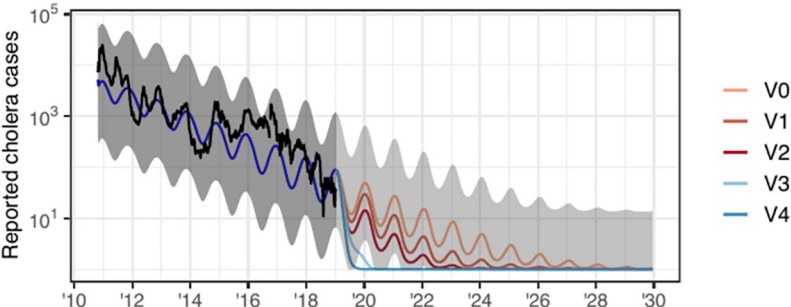

**Fig 5. Simulated trajectory of Model 2.** The black line shows the nationally aggregated weekly cholera incidence data. The blue curve from 2012–2019 is the trajectory of the calibrated version of Model 2. Projections under the various vaccination scenarios, which are discussed in detail in the **Forecasts** subsection are also included. The gray ribbons represent a 95% interval obtained from the log-normal measurement model. To avoid over-plotting, measurement variance is only plotted for the V0 vaccination scenario.

environmental, or extra-demographic) which is dominant at high population counts. Here we chose to fit the model to transformed case counts rather than adding overdispersion to the measurement model with the goal of minimizing the changes to the model proposed by Lee et al. [4].

We implemented this model using the `spatPomp` R package [93]. The model was then fit using the subplex algorithm [94]. A comparison of the trajectory of the fitted model to the data is given in Fig 5.

**Calibrating Model 3 parameters.** Model 3 describes cholera dynamics in Haiti using a metapopulation model, where the hidden states in each administrative department has an effect on the dynamics in other departments. The decision to address metapopulation dynamics using a spatially explicit model, rather than to aggregate over space, is double-edged. Evidence for the former approach has been provided in previous studies [95], including the specific case of heterogeneity between Haitian departments in cholera transmission [19]. However, a legitimate preference for simplicity can support a decision to consider nationally aggregated models [49, 96].

In our literature review, 17 articles considered dynamic models that incorporate spatial heterogeneity [4, 5, 8–11, 14, 17, 19, 20, 22–25, 27, 33, 34]. All but four [4, 27, 33, 34] of these studies used deterministic dynamic models: this greatly simplifies the process of calibrating model parameters to incidence data, though deterministic models can struggle to describe complex stochastic dynamics. The model in [27] was fit using an Ensemble Kalman Filter (EnKF) [97]; though EnKF scales favorably with the number of spatial units, it relies on linearization of latent states which can be problematic for highly nonlinear systems [98, 99]. Alternative approaches used to fit stochastic models included making additional simplifying assumptions to aid in the fitting process [4], and using MCMC algorithms [33, 34] which require specific structures in the latent dynamics, making these algorithms non plug-and-play. In this subsection, we present how the recently developed iterated block particle filter (IBPF) algorithm [100, 101] can be used to fit a spatially explicit stochastic dynamic model to incidence data.

One issue that arises when fitting spatially explicit models is that parameter estimation techniques based on the particle filter become computationally intractable as the number of spatial units increases. This is a result of the approximation error of particle filters growing exponentially in the dimension of the model [102, 103]. To avoid the approximation error present in high-dimensional models, Lee et al. [4] simplified the problem of estimating the parameters of Model 3 by creating an approximate version of the model where the units are independent

given the observed data. Reducing a spatially coupled model to individual units in this fashion requires special treatment of any interactive mechanisms between spatial units, such as found in Eq (25). Because the simplified, spatially-decoupled version of Model 3 implemented in [4] relies on the observed cholera cases, the calibrated model cannot readily be used to obtain forecasts. Therefore, in order to obtain model forecasts, Lee et al. [4] used the parameters estimates from the spatially-decoupled approximation of Model 3 to obtain forecasts using the fully coupled version of the model. This approach of model calibration and forecasting avoids the issue of particle depletion, but may also be problematic. One concern is that cholera dynamics in department $u$ are highly related to the dynamics in the remaining departments; calibrating model parameters while conditioning on the observed cases in other departments may therefore lead to an over-dependence on observed cholera cases. Another concern is that the two versions of the model are not the same, resulting in sub-optimal parameter estimates for the spatially coupled model, as parameters that maximize the likelihood of the decoupled model almost certainly do not maximize the likelihood of the fully coupled model. These two concerns may explain the unrealistic forecasts and low likelihood of Model 3 in [4] (Table 1).

At the time Lee et al. [4] conducted their study, there was no known algorithm that could readily be used to maximize the likelihood of an arbitrary meta-population POMP model with coupled spatial dynamics, which justifies the spatial decoupling approximation that was used to calibrate model parameters. For our analysis, we calibrate the parameters of the spatially coupled version of Model 3 using the IBPF algorithm [101]. This algorithm extends the work of Ning and Ionides [100], who provided theoretic justification for the version of the algorithm that only estimates unit-specific parameters. The IBPF algorithm enables us to directly estimate the parameters of models describing high-dimensional partially-observed nonlinear dynamic systems via likelihood maximization. The ability to directly estimate parameters of Model 3 is responsible for the large increase in model likelihoods reported in Table 1. Simulations from the fitted model are displayed in Fig 6.

## Model diagnostics

The goal of parameter calibration—whether done using Bayesian or frequentist methods—is to find the best description of the observed data in the context of the model. Obtaining the best fitting set of parameters for a given model does not, however, guarantee that the model provides an accurate representation of the system under investigation. Model misspecification, which may be thought of as the omission of a mechanism in the model that is an important

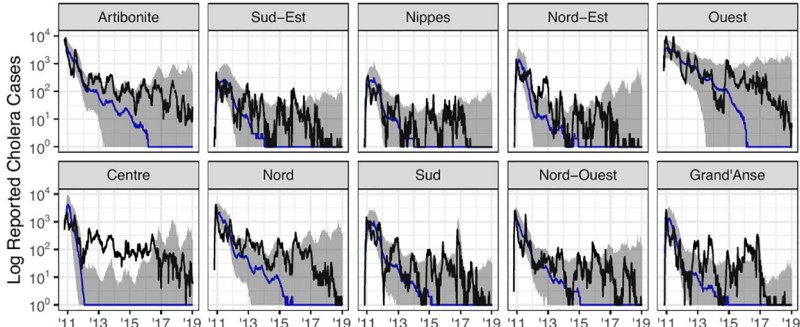

**Fig 6. Simulations from Model 3 compared to reported cholera cases.** Simulations from initial conditions using the spatially coupled version of Model 3. The black curve represents true case count, the blue line the median of 500 simulations from the model, and the gray ribbons representing 95% confidence interval.

feature of the dynamic system, is inevitable at all levels of model complexity. To make progress, while accepting proper limitations, one must bear in mind the much-quoted observation of George Box [104] that "all models are wrong but some are useful." Beyond being good practical advice for applied statistics, this assertion is relevant for the philosophical justification of statistical inference as severe testing [105]. In this section, we discuss some tools for diagnosing mechanistic models with the goal of making the subjective assessment of model "usefulness" more objective. To do this, we will rely on the quantitative ability of the model to match the observed data, which we call the model's *goodness-of-fit*, with the guiding principle that a model which cannot adequately describe observed data may not be reliable for useful purposes. Goodness-of-fit may provide evidence supporting the causal interpretation of one model versus another, but cannot by itself rule out the possibility of alternative explanations.

One common approach to assess a mechanistic model's goodness-of-fit is to compare simulations from the fitted model to the observed data. Visual inspection may indicate defects in the model, or may suggest that the observed data are a plausible realization of the fitted model. While visual comparisons can be informative, they provide only a weak and informal measure of the goodness-of-fit of a model. The study by Lee et al. [4] provides an example of this: their models and parameter estimates resulted in simulations that visually resembled the observed data, yet resulted in model likelihoods that were considerably smaller than likelihoods that can be achieved (see Table 1). Alternative forms of model validation should therefore be used in conjunction with visual comparisons of simulations to observed data.

Another approach is to compare a quantitative measure of the model fit (such as MSE, predictive accuracy, or model likelihood) among all proposed models. These comparisons, which provide insight into how each model performs relative to the others, are quite common [10, 33]. To calibrate relative measures of fit, it is useful to compare against a model that has well-understood statistical ability to fit data, and we call this model a *benchmark*. Standard statistical models, interpreted as associative models without requiring any mechanistic interpretation of their parameters, provide suitable benchmarks. Examples include linear regression, auto regressive moving average (ARMA) time series models, or even independent and identically distributed measurements. Benchmarks enable us to evaluate the goodness of fit that can be expected of a suitable mechanistic model.

Associative models are not constrained to have a causal interpretation, and typically are designed with the sole goal of providing a statistical fit to data. Therefore, we should not require a candidate mechanistic model to beat all benchmarks. However, a mechanistic model which falls far short against benchmarks is evidently failing to explain some substantial aspect of the data. A convenient measure of fit should have interpretable differences that help to operationalize the meaning of far short. Ideally, the measure should also have favorable theoretical properties. Consequently, we focus on log-likelihood as a measure of goodness of fit, and we adjust for the degrees of freedom of the models to be compared by using the Akaike information criterion (AIC) [106].

In some cases, a possible benchmark model could be a generally accepted mechanistic model, but often no such model is available. Because of this, we use a simple negative binomial model with an auto regressive mean as our associative benchmark; this model is described in (35).

$$Y_n | Y_{n-1} \sim \text{NB}(\alpha + \beta Y_{n-1}, \varphi), \tag{35}$$

where $\text{E}(Y_n|Y_{n-1}) = \alpha + \beta Y_{n-1}$, and $\text{Var}(Y_n|Y_{n-1}) = \text{E}(Y_n|Y_{n-1}) + \text{E}(Y_n|Y_{n-1})^2/\varphi$. To obtain a benchmark for models with a meta-population structure, we fit independent auto-regressive negative binomial models to each spatial unit. Under the assumption of independence, the

log-likelihood of the benchmark on the entire collection of data can be obtained by summing up the log-likelihood for each independent model. In general, a spatially explicit model may not have well-defined individual log-likelihoods, and, in this case, comparisons to benchmarks must be made at the level of the joint model.

In the case where the case counts are large, an alternative benchmark recommended by He et al. [2] is a log-linear Gaussian ARMA model; the theory and practice of ARMA models is well developed, and these linear models are appropriate on a log scale due to the exponential growth and decay characteristic of biological dynamics. We use the auto regressive negative binomial model, however, because the large number of weeks with zero recorded cholera cases in department level data makes a benchmark based on a continuous distribution problematic. Log-likelihoods and AIC values of Models 1–3 and of their respective benchmark models are provided in Table 1. Models that are fit to the same datasets can be directly compared using AIC values, making it a useful tool to compare to benchmark models. Though Models 2 and 3 are both fit to department level incidence reports, their AIC values are not directly comparable due to the way Model 3 initializes latent states (S2 Text).

It should be universal practice to present measures of goodness of fit for published models, and mechanistic models should be compared against benchmarks. In our literature review of the Haiti cholera epidemic, no non-mechanistic benchmark models were considered in any of the 32 papers that used dynamic models to describe cholera in order to obtain scientific conclusions. Including benchmarks would help authors and readers to detect and confront any major statistical limitations of the proposed mechanistic models. In addition, the published goodness of fit provides a concrete point of comparison for subsequent scientific investigations. When combined with online availability of data and code, objective measures of fit provide a powerful tool to accelerate scientific progress, following the paradigm of the *common task framework* [107].

The use of benchmarks may also be beneficial when developing models at differing spatial scales, where a direct comparison between model likelihoods is meaningless. In such a case, a benchmark model can be fit to each spatial resolution being considered, and each model compared to their respective benchmark. Large advantages (or shortcomings) in model likelihood relative to the benchmark for a given spatial scale that are not present in other spatial scales may provide weak evidence for (or against) the statistical fit of models across a range of spatial resolutions.

Comparing model log-likelihoods to a suitable benchmark may not be sufficient to identify all the strengths and weaknesses of a given model. Additional techniques include the inspection of conditional log-likelihoods of each observation given the previous observations in order to understand how well the model describes each data point (S5 Text). Other tools include plotting the effective sample size of each observation [108]; plotting the values of the hidden states from simulations (S5 Text); and comparing summary statistics of the observed data to simulations from the model [95, 109].

## Corroborating fitted models with scientific knowledge

The resulting mechanisms in a fitted model can be compared to current scientific knowledge about a system. Agreement between model-based inference and our current understanding of a system may be taken as a confirmation of both model-based conclusions and our scientific understanding. On the other hand, comparisons may generate unexpected results that have the potential to spark new scientific knowledge [110].

In the context of our case study, we demonstrate how the fit of Model 1 corroborates other evidence concerning the role of rainfall in cholera epidemics. Specifically, we examine the

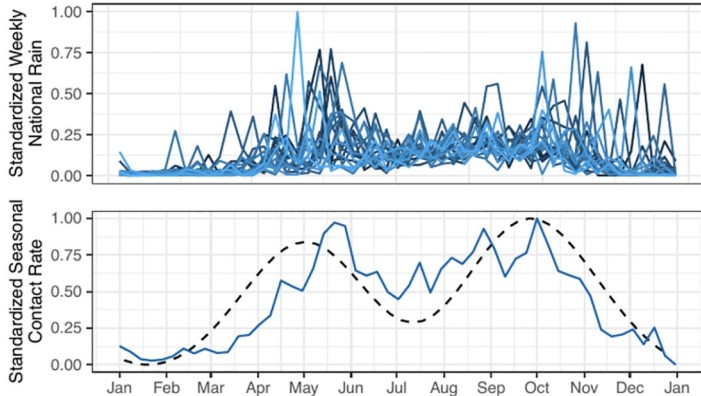

**Fig 7. Seasonality of Model 1 transmission compared to rainfall data.** (Top) weekly rainfall in Haiti, lighter colors representing more recent years. (Bottom) estimated seasonality in the transmission rate (dashed line) plotted alongside mean rainfall (solid line). The outsized effect of rainfall in the fall may be due to Hurricane Matthew, which struck Haiti in October of 2016 and resulted in an increase of cholera cases in the nation.

results of fitting the flexible cubic spline term in Model 1 (Eqs (3) and (4)). The cubic splines permit flexible estimation of seasonality in the force of infection, $\beta(t)$. Fig 7 shows that the estimated seasonal transmission rate $\beta$ mimics the rainfall dynamics in Haiti, despite Model 1 not having access to rainfall data. This is consistent with previous studies that incorporated rainfall as an important part of their mechanistic model or otherwise argue that rainfall is an important driver of cholera dynamics in Haiti [4, 9, 10, 18, 26, 27, 29, 63, 64]. The estimated seasonality also features an increased transmission rate during the fall, which was noticed at an earlier stage of the epidemic [10]. The high transmission rate in the fall may be a result of the increase transmission that occurred in the fall of 2016, when hurricane Matthew struck Haiti [77].

For any model-based inference, it is important to recognize and assess the modeling simplifications and assumptions that were used in order to arrive at the conclusions. In epidemiological studies, for example, quantitative understanding of individual-level processes may not perfectly match model parameters that were fit to population-level case counts, even when the model provides a strong statistical fit [2]. This makes direct interpretation of estimated parameters delicate.

Our case study provides an example of this in the parameter estimate for the duration of natural immunity due to cholera infection, $\mu_{RS}^{-1}$. Under the framework of Model 2, the best estimate for this parameter is $1.4 \times 10^{11}$ yr, suggesting that individuals have permanent immunity to cholera once infected. Rather than interpreting this as scientific evidence that individuals have permanent immunity from cholera, this result suggests that Model 2 favors a regime where reinfection events are a negligible part of the dynamics. The depletion of susceptible individuals may be attributed to confounding mechanisms—such as localized vaccination programs and non-pharmaceutical interventions that reduce cholera transmission [17, 83]—that were not accounted for in the model. Perhaps the best interpretation of the estimated parameter, then, is that under the modeling framework that was used, the model most adequately describes the observed data by having a steady decrease in the number of susceptible individuals. The weak statistical fit of Model 2 compared to a log-linear benchmark (see Table 1) cautions us against drawing quantitative conclusions from this model. A model that has a poor statistical fit may nevertheless provide a useful conceptual framework for thinking about the system under investigation. However, a claim that the model has been validated against data

should be reserved for situations where the model provides a statistical fit that is competitive against alternative explanations.

A model which aspires to provide quantitative guidance for assessing interventions should provide a quantitative statistical fit for available data. However, strong statistical fit does not guarantee a correct causal structure: it does not even necessarily require the model to assert a causal explanation. A causal interpretation is strengthened by corroborative evidence. For example, reconstructed latent variables (such as numbers of susceptible and recovered individuals) should make sense in the context of alternative measurements of these variables [111]. Similarly, parameters that have been calibrated to data should make sense in the context of alternative lines of evidence about the phenomena being modeled, while making allowance for the possibility that the interpretations of parameters may vary when modeling across differing spatial scales.

In the supplement material (S5 Text), we explore in more detail the process of model fitting and diagnostics for Model 3. Here we demonstrate that the model outperforms its benchmark model on the aggregate scale. However, when focusing on the spatial units with the highest incidence of cholera, Model 3 performs roughly the same as a simple benchmark. By comparing simulations from the fitted model to the filtering distribution, we see that the reconstructed latent states of the model favor higher levels of cholera transmission than what is typically observed in the incidence data. These results hint at the possibility of model misspecification, and warrant a degree of caution in interpreting the model's outputs.

## Results

### Forecasts

Forecasts are an attempt to provide an accurate estimate of the future state of a system based on currently available data, together with an assessment of uncertainty. Forecasts from mechanistic models that are compatible with current scientific understanding may also provide estimates of the future effects of potential interventions. Further, they may enable real-time testing of new scientific hypotheses [112].

Forecasts of a dynamic system should should be consistent with the available data. It is particularly important that forecasts are consistent with the most recent information available, as recent data is likely to be more relevant than older data. While this assertion may seem self-evident, it is not the case for deterministic models, for which the initial conditions together with the parameters are sufficient for forecasting, and so recent data may not be consistent with model trajectories. Epidemiological forecasts based on deterministic models are not uncommon in practice, despite their limitations [95]. Lee et al. [4] chose to obtain forecasts from all of their models by simulating forward from initial conditions, rather than conditioning forecasts based on the available data. This decision is possibly as a result of using a deterministic model, as forecasts from different models may only be considered comparable if they are obtained in the same way, which is most easily done be by simulating from initial conditions because Model 2 is deterministic.

In contrast, for non-deterministic Models 1 and 3, we obtain forecasts by simulating future values using latent states that are harmonious with the most recent data. This is done by simulating forward from latent states drawn at the last observation time ($t_N$) from the filtering distribution $f_{X_N|Y_{1:N}}(\boldsymbol{x}_N|\boldsymbol{y}_{1:N}^*; \hat{\theta})$. The decision to obtain model forecasts from initial conditions partially explains the unsuccessful forecasts of Lee et al. [4]. Table S7 in their supplement material, which contains results that were not discussed in their main article, shows that the subset of their simulations with zero cholera cases from 2019–2020 also correspond with its

disappearance until 2022. These results support our argument that forecasts should be made by ensuring the starting point for the forecast is consistent with available data.

Uncertainty in just a single parameter can lead to drastically different forecasts [49]. Therefore, parameter uncertainty should also be considered when obtaining model forecasts to influence policy. If a Bayesian technique is used for parameter estimation, a natural way to account for parameter uncertainty is to obtain simulations from the model where each simulation is obtained using parameters drawn from the estimated posterior distribution. For frequentist inference, one possible approach is obtaining model forecasts from various values of $\theta$, where the values of $\theta$ are sampled proportionally according to their corresponding likelihoods [95] (S6 Text). Both of these approaches share the similarity that parameters are chosen for the forecast approximately in proportion to their corresponding value of the likelihood function, $f_{Y_{1:N}}(y_{1:N}^*; \theta)$. In this analysis, we do not construct forecasts accounting for parameter uncertainty as our focus is on the estimation and diagnosis of mechanistic models, rather than providing forecasts intended to influence policy. Furthermore, we use the projections from a single point estimate to highlight the deficiency of deterministic models that the only variability in model projections is a result of parameter and measurement uncertainty, which can lead to over-confidence in forecasts [95].

The primary forecasting goal of Lee et al. [4] was to investigate the potential consequences of vaccination interventions on a system to inform policy. One outcome of their study include estimates for the probability of cholera elimination under several possible vaccination scenarios. Mimicking their approach, we define cholera elimination as an absence of cholera infections for at least 52 consecutive weeks, and we provide forecasts under the following vaccination scenarios:

$V0$:   No additional vaccines are administered.

$V1$:   Vaccination limited to the departments of Centre and Artibonite, deployed over a two-year period.

$V2$:   Vaccination limited to three departments: Artibonite, Centre, and Ouest deployed over a two-year period.

$V3$:   Countrywide vaccination implemented over a five-year period.

$V4$:   Countrywide vaccination implemented over a two-year period.

Simulations from probabilistic models (Models 1 and 3) represent possible trajectories of the dynamic system under the scientific assumptions of the models. Because Model 1 only accounts for national level disease dynamics, the pre-determined department-specific vaccination campaigns are carried out by assuming the vaccines are administered in one week to the same number of individuals that would have obtained vaccines if explicitly administered to the specific departments. We refer readers to [4] and the accompanying supplement material for more details. Estimates of the probability of cholera elimination can therefore be obtained as the proportion of simulations from these models that result in cholera elimination. The results of these projections are summarized in Fig 8.

Probability of elimination estimates of this form are not meaningful for deterministic models, as the trajectory of these models only represent the mean behavior of the system rather than individual potential outcomes. We therefore do not provide probability of elimination estimates under Model 2, but show trajectories under the various vaccination scenarios using this model (Fig 5).

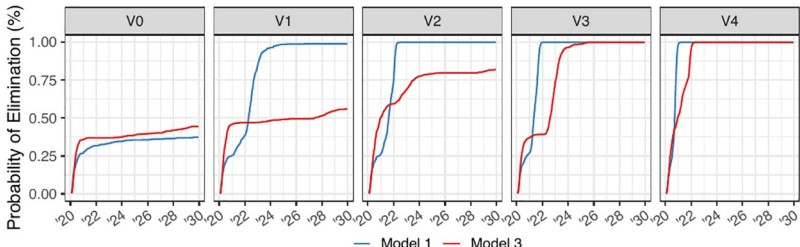

**Fig 8. Simulated probability of elimination using Models 1 and 3.** Probability of cholera elimination, defined as having zero cholera infectious for at least 52 consecutive weeks, based on 10 year simulations from calibrated versions of Models 1 and 3. Compare to Fig 3A of [4].

## Discussion

The ongoing global COVID-19 pandemic has demonstrated how government policy may be affected by the inferences drawn from mathematical modeling [49]. However, the development of credible models—which are supported by data and can provide quantitative insights into a dynamic system—remains a challenging task. In this article, we demonstrated opportunities available for raising the current standards of statistical inference for mathematical models of biological systems.

We presented methodology consistent with existing guidelines [48] but going beyond standard practice. In particular, we showed the value of comparing the likelihood of fitted mechanistic models versus non-mechanistic benchmarks, a practice that has been previously advocated for [2] but was not done by any of the studies in our literature review. These comparisons, along with other likelihood based diagnostics, help identify specific limitations of proposed models. Diagnostic tools include likelihood profile methods, which help to assess parameter identifiability and enable the construction of confidence intervals for parameter estimates [84, 113]. When reaching conclusions, it is important to consider potential consequences of confounded variables and model misspecification.

Model diagnostics are a key tool for exposing unresolved model limitations and improving model fit. In our case study, we compared the three models from Lee et al. [4] to statistical benchmarks, revealing areas for improvement. For example, comparisons of Model 3 to a benchmark revealed its inadequacy in accounting for the post-hurricane increase in transmission, leading to a beneficial model refinement. When a mechanistic model is competitive with statistical benchmarks, we have a license to begin critical evaluation of its causal implications. If a model falls far behind simple benchmarks, there is likely to be substantial limitation in the data analysis that should be identified and remedied. In our case study, the re-calibrated version of Model 1 outperformed its benchmark, so we proceeded to examine causal implications. When doing so, we found that the fitted model provides a causal description of the dynamic system that is consistent with known features of the system, such as the importance of rainfall as a driver of cholera infection. The congruency between causal implications of the model and our belief about the dynamic system, coupled with a strong quantitative description of observed data relative to a benchmark, provides support for viewing the model as a plausible quantitative representation of the system under investigation.

When fitting a mechanistic model to a dynamic system, the complexity of the model warrants consideration. Mathematical models provide simplified representations of complex systems, with the simplicity serving both to facilitate scientific understanding and to enable statistical inference on unknown parameters. In our case study, employing deterministic

dynamics in Model 2 was found to be an over-simplification by comparing model fit with benchmarks. Model 3 is distinct in that it is both stochastic and has a meta-population structure, making it challenging to draw likelihood-based inferences. In this paper, we demonstrated how this model class can be calibrated to incidence data using the innovative IBPF algorithm. One of only a few examples of fitting a nonlinear non-Gaussian meta-population model via maximum likelihood [55, 101], this case study exemplifies the algorithm's potential benefits and provides an example for future researchers on a possible approach to fitting a high-dimensional non-linear model.

Likelihood-based methods aid in determining an appropriate level of model complexity. Models fit to the same data can be compared using a criteria such as AIC. Nested model variations are particularly useful as they enable formal statistical testing of the nested features via likelihood ratio tests. Our case study demonstrated the examination of nested model features for all three models. Model 1 investigated a time-varying transmission rate; Model 2 assessed a phase-shift parameter in seasonal cholera peaks; Model 3 incorporated hurricane-related parameters.

Unmodeled features of a dynamic system can lead to spurious or misleading parameter estimates if the features substantially impact observed data. In deterministic models, features that cannot be explained by measurement error must be accounted for by the choice of parameters. For our case study, some of the parameter estimates for the deterministic Model 2 are implausible, such as the infinite immunity discussed above, and this may be explained by compensation for model misspecification. Incorporating demographic and environmental stochasticity into models can mitigate the impact of unmodeled features. Stochastic phenomena are not only arguably present in biological systems, but their inclusion in a model also allows observed data variations to be attributed to inherent uncertainty rather than to distorted parameter values. Models 1 and 3 suggest the presence of extra-demographic stochasticity [2, 55, 76], as evidenced by the confidence intervals for the corresponding parameter $\sigma_{\mathrm{proc}}$ (S7 Text).

If forecasts are an important component of a modeling task, the forecasts should be consistent with the available data, particularly at the most recently available time points. In our case study, we did this by simulating forward from the filtering distribution, as this procedure conditions latent variables on the available data. This type of forecasting, however, is not directly available using a deterministic model, where future dynamics are fully determined by initial conditions and parameter values. This can result in over-confident model forecasts [95]. Despite their limitations, deterministic models can offer valuable insights into dynamic systems [58]. In [4], the forecasts from the deterministic Model 2 were qualitatively more consistent with the observed disappearance of cholera than the stochastic models. In our case study, we found improvements to Models 1 and 3 that resulted in improved forecasts for these models.

In our case study, we found that additional attention to statistical details could have resulted in an enhanced statistical fit to the observed incidence data. This would have improved the accuracy of the policy guidance resulting from the study. We used the same data, models, and much of the same code used by Lee et al. [4], but we arrived at drastically different conclusions. Specifically, each of the re-calibrated models predicted with moderate probability that cholera would disappear from Haiti. Although there have been new cases of cholera in Haiti, this conclusion aligns more with the prolonged absence of cholera cases from 2019–2022. We acknowledge the benefit of hindsight: our demonstration of a statistically principled route to obtain better-fitting models resulting in more robust insights does not rule out the possibility of discovering other models that fit well yet predict poorly.

Mechanistic models offer opportunities for understanding and controlling complex dynamic systems. This case study has investigated issues requiring attention when applying

powerful new statistical techniques that can enable statistically efficient inference for a general class of partially observed Markov process models. Researchers should ensure that intensive numerical calculations are adequately executed. Using benchmarks and alternative model specifications to assess statistical goodness-of-fit should also should be common practice. Once a model has been adequately calibrated to data, care is required to assess what causal conclusions can properly be inferred given the possibility of alternative explanations consistent with the data. Studies that combine model development with thoughtful data analysis, supported by a high standard of reproducibility, build knowledge about the system under investigation. Cautionary warnings about the difficulties inherent in understanding complex systems [49, 110, 114] should motivate us to follow best practices in data analysis, rather than avoiding the challenge.

## Reproducibility and extendability

Lee et al. [4] published their code and data online, and this reproducibility facilitated our work. Robust data analysis requires not only reproducibility but also extendability: if one wishes to try new model variations, or new approaches to fitting the existing models, or plotting the results in a different way, this should not be excessively burdensome. Scientific results are only trustworthy so far as they can be critically questioned, and an extendable analysis should facilitate such examination [115].

We provide a strong form of reproducibility, as well as extendability, by developing our analysis in the context of a software package, `haitipkg`, written in the R language [116]. Using a software package mechanism supports documentation, standardization and portability that promote extendability. In the terminology of Gentleman and Temple Lang [115], the source code for this article is a *dynamic document* combining code chunks with text. In addition to reproducing the article, the code can be extended to examine alternative analysis to that presented. The dynamic document, together with the R packages, form a *compendium*, defined by Gentleman and Temple Lang [115] as a distributable and executable unit which combines data, text and auxiliary software (the latter meaning code written to run in a general-purpose, portable programming environment, which in this case is R).

## Supporting information

**S1 Fig. Model 1.** Flow chart representation of Model 1.
(PDF)

**S2 Fig. Model 2.** Flow chart representation of Model 2.
(PDF)

**S3 Fig. Model 3.** Flow chart representation of Model 3.
(PDF)

**S1 Table. Notation conversion table.** Conversions between the notation used here and the notation of Lee et al. [4].
(PDF)

**S1 Text. Model details.** In depth description of the Markov chain and differential equation interpretations of compartment flow rates, as well as details on the numeric implementation of these models.
(PDF)

**S2 Text. Initialization models.** Additional details of the initialization model that were used for Models 1–3.
(PDF)

**S3 Text. Measurement models.** Additional details of the measurement model that were used for Models 1–3.
(PDF)

**S4 Text. Likelihood of Lee et al. [4] models.** Details of how the log-likelihood of the Lee et al. [4] models were calculated.
(PDF)

**S5 Text. Calibrating Model 3.** Additional details on the procedures for fitting and diagnosing Model 3.
(PDF)

**S6 Text. Accounting for parameter uncertainty.** A description of a empirical-Bayes approach that can be used to account for parameter uncertainty.
(PDF)

**S7 Text. Confidence intervals.** Tables and figures describing 95% confidence intervals for estimated parameters.
(PDF)

## Acknowledgments

The authors would like to thank Mercedes Pascual and Betz Halloran for helpful discussions. Laura Matrajt provided additional data for the Model 2 analysis.

## Author Contributions

**Conceptualization:** Edward L. Ionides.

**Formal analysis:** Jesse Wheeler, AnnaElaine Rosengart, Zhuoxun Jiang.

**Funding acquisition:** Edward L. Ionides.

**Investigation:** Jesse Wheeler, AnnaElaine Rosengart, Zhuoxun Jiang, Kevin Tan, Noah Treutle.

**Methodology:** Jesse Wheeler, Edward L. Ionides.

**Project administration:** Jesse Wheeler.

**Software:** Jesse Wheeler, AnnaElaine Rosengart, Zhuoxun Jiang.

**Supervision:** Edward L. Ionides.

**Validation:** Jesse Wheeler.

**Visualization:** Jesse Wheeler, AnnaElaine Rosengart, Kevin Tan.

**Writing – original draft:** Jesse Wheeler, Edward L. Ionides.

**Writing – review & editing:** Jesse Wheeler, AnnaElaine Rosengart, Zhuoxun Jiang, Kevin Tan, Edward L. Ionides.

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
