## [Decision Letter · Decision Letter 0]

8 Jan 2024

Dear Mr Wheeler,

Thank you very much for submitting your manuscript "Informing policy via dynamic models: Cholera in Haiti" for consideration at PLOS Computational Biology.

As with all papers reviewed by the journal, your manuscript was reviewed by members of the editorial board and by several independent reviewers. In light of the reviews (below this email), we would like to invite the resubmission of a significantly-revised version that takes into account the reviewers' comments.

The paper has been read by two experts in the field, and briefly by me. We are all in agreement that the paper is interesting, but also that it requires some work before being publishable in PLoS Comp Biol. In particular, it is important to explain the pedagogical purpose of paper. Both referee reports are very well written with several constructive suggestions. If you revise the manuscript according to the suggestions, the manuscript has a high chance of being accepted.

Kind regards, Tom Britton

We cannot make any decision about publication until we have seen the revised manuscript and your response to the reviewers' comments. Your revised manuscript is also likely to be sent to reviewers for further evaluation.

Sincerely,

Tom Britton

Academic Editor

PLOS Computational Biology

Virginia Pitzer

Section Editor

PLOS Computational Biology

The paper has been read by two experts in the field, and briefly by me. We are all in agreement that the paper is interesting, but also that it requires some work before being publishable in PLoS Comp Biol. In particular, it is important to explain the pedagogical purpose of paper. Both referee reports are very well written with several constructive suggestions. If you revise the manuscript according to the suggestions, the manuscript has a high chance of being accepted.

Kind regards, Tom Britton

Reviewer's Responses to Questions

**Comments to the Authors:**

Reviewer #1: Review uploaded as an attachment

Reviewer #2: General comments:

This well-written article is a strong piece of work that will be useful to the readers of this journal and, in particular, to researchers who perform statistical inference on compartmental models of infectious diseases. Furthermore, this paper is enhanced by its transparency and reproducibility, as the provided code is well-documented and organised in an R package. Additionally, the supplementary information is a valuable resource for researchers in this field.

In a nutshell, the authors argue that existing criteria to evaluate the validity of a disease model are insufficient. Therefore, they propose more stringent standards for evaluating models’ ability to fit the available data in order to obtain more reliable forecasts. The authors use a Cholera case study to outline their suggestions. To highlight, a key contribution from this work is the recommendation of employing inductive (associative) models as a goodness-of-fit benchmark, as evidenced by this sentence: “It should be universal practice to present measures of goodness of fit for published models, and mechanistic models should be compared against benchmarks”. Undoubtedly, this approach provides an objective measure to judge the ability of mechanistic models to fit the data. In the following sections, I express my opinion on how this paper may be improved.

Major Concerns

1. Literature

While the arguments provided throughout the article are well-articulated, there needs to be more supporting literature at the beginning of major sections. For instance, I don’t need to be convinced that the structure of a model should be based on a realistic theory about the observed phenomenon; namely, models should be a white box. However, not everyone is on board with this premise, and supporting literature that argues in favour of this approach should be mentioned. In short, more citations should be added at the beginning of each major section.

2. Limitations.

In various passages of this paper, it is hinted that modellers should refrain from deterministic models and instead opt for more realistic stochastic representations. While the critique of ODE models is valid, the shift to stochastic structures is not a free choice. For example, introducing extra-demographic variability adds one additional parameter (infinitesimal variance). In ODE models, one extra parameter can lead to unidentifiability, and there's no apparent reason why this would differ in a stochastic version. Moreover, transitioning to stochastic models involves abandoning well-established MCMC algorithms in favour of methods still in development. Therefore, modellers should not assume that more realistic models with additional parameters are necessarily better without proper caution. In my experience, diagnosing unidentifiability is easier in ODE models than in POMP structures. Hence, there is a trade-off between benefits and costs.

Moreover, Monte Carlo methods, such as the Particle Filter and, by extension, Iterated Filtering, aim to approximate integrals (the posterior or filtering distributions). However, one cannot take for granted that these methods provide accurate descriptions of these targets without proper validation. In more ‘traditional’ MCMC methods, diagnostics like the potential scale reduction factor and effective sample size play a crucial role in the inference process. Unsatisfactory values of these diagnostics render inferences unreliable, often necessitating model reformulation. In contrast, in the literature on POMP models (including this paper), diagnostics are tangentially mentioned. Users (like me) sometimes face uncertainty about whether the lack of fit is due to model misspecification or problems with the Monte Carlo algorithm exploring the parameter space.

For example, in Figure 5, department Ouest exhibits substantial uncertainty from 2014 onwards, and this figure is on a log scale. Essentially, the inference suggests that 'anything can happen'. I would like to pinpoint the nature of this collapse in uncertainty. Identifiability issues might be at play, given the possibility of more estimated parameters than the incidence data can inform. Sometimes, we ask too much from the data. Observe the discrepancy in trends between the average behaviour and the uncertainty ribbons.

In summary, the authors should elaborate on the limitations of the proposed approach.

3. Conclusions

I find that the conclusions are somewhat disconnected from the introduction and abstract, which state that the paper presents a methodology to diagnose model misspecification, develop alternative models, and make computational improvements. It would improve readability to include a summary in this section. Specifically, link each contribution to a particular example. For instance, in the case of model 1, computational improvements increased the log-likelihood. In short, connect the findings more explicitly to the research question and stated goals.

Minor comments:

1. In the author summary, this part is hard to follow: “and provides careful justification of valid conclusions from the fitted model. Objective measures are used to benchmark model fit; when these are combined with reproducibility, a framework emerges for continual improvement when revisiting the data and models.” Please rephrase.

2. Lines 1-8. Please add more citations.

3. Line 36. Can you be explicit about what the forecasts predicted? Did the models predict a rise in cases?

4. Line 42. Add hyphen: “Model-based conclusions”.

5. Lines 67-78. Add more citations.

6. Please add uncertainty intervals to the estimated parameters in Table 1. If necessary, consider splitting the table into two or including this additional information in the supplementary material. I suggest this update because there is an indication of unidentifiability when parameter estimates are fairly broad.

7. In line 159, it is stated that “\\upsilon^\\star(t) is efficacy at time t since vaccination for adults” and then “single and double vaccine doses were modeled by changing the waning of protection; protection was modeled as equal between single and double dose until 52 weeks after vaccination, at which point the single dose becomes ineffective”. I examined the reference from which this function is based but found only a numeric table. Please provide the equation of this function or a detailed description in the supplementary information. It would be beneficial for readers to understand how to model this complex feature.

8. Lines 209-211. In model 2, an incidence measurement is employed to configure a prevalence compartment. As the authors may know, initial values severely condition the dynamics of a model. Please explain this decision. Is it because that was the approach followed in the original formulation (Lee at al's paper)?

9. Lines 245-247. Same comment as before. What’s the justification for assuming incidence measurements as the basis for prevalence states? What are the risks?

10. Line 262. “deterministic Model 2 is a degenerate case of a stochastic model”. Please explain why or provide a reference.

11. Lines 260-274. Please add more citations.

12. Lines 343-345. This sentence is key for this paper, but it’s hard to follow: “The robust statistical conclusion is that a model which allows for change fits better than one which does not—we argue that a decreasing transmission rate is a plausible way to explain this, but the incidence data themselves do not provide enough information to pin down the mechanism”. Please rephrase.

13. Line 357. ”Determining the necessary computational effort needed to maximize model likelihoods and acting accordingly” How do we determine the necessary computational effort?

14. Please add the predicted intervals to Fig 4. I would like to see the effect of the log-normal measurement model.

15. Lines 513-515. Please clarify the comparison between disaggregated models and a benchmark. Let’s say I have spatial units 1 and 2, for which I have observations y1 and y2. Should I fit the disaggregated mechanistic model to y1 and y2 simultaneously (as usual) but keep a record of the individual log-likelihoods (log-lik y1 & log-lik y2). In parallel, fit the benchmark independently to y1 and y2, and then compare by log-likelihoods or information criteria by spatial unit.

16. Line 528. Please add hyphen: “Model-based inference”.

17. Lines 576-577. “We notice that the calibrated model favors higher levels of cholera transmission than what was typically observed in the incidence data (S5 Text)”. After this fragment, please summarise in one or two sentences what it will be found in S5 Text.

18. Lines 599-601. “The decision not to do this partially explains the unsuccessful forecasts of Lee et al. [1]: their Table S7 shows that the subset of their simulations which were consistent with observing zero cases in 2019 also accurately predicted the prolonged absence of detected cholera”. The fragment before the colon says that Lee was unsuccessful. However, the fragment after the colon says that was in part successful. Please clarify.

19. Line 631. Since Model 1 accounts for infections at the national level, how are scenarios V1 and V2 handled?

**Have the authors made all data and (if applicable) computational code underlying the findings in their manuscript fully available?**

Reviewer #1: Yes

Reviewer #2: Yes

PLOS authors have the option to publish the peer review history of their article (what does this mean?). If published, this will include your full peer review and any attached files.

Reviewer #1: No

Reviewer #2: No

Figure Files:

Data Requirements:

Reproducibility:

To enhance the reproducibility of your results, we r

---

## [Decision Letter · Decision Letter 1]

29 Mar 2024

Dear Mr Wheeler,

We are pleased to inform you that your manuscript 'Informing policy via dynamic models: Cholera in Haiti' has been provisionally accepted for publication in PLOS Computational Biology.

Best regards,

Tom Britton

Academic Editor

PLOS Computational Biology

Virginia Pitzer

Section Editor

PLOS Computational Biology

The reviewers are happy with the revision and so I am. For this reason my recommendation is accept the paper for publication.

Kind regards, Tom Britton

Reviewer's Responses to Questions

**Comments to the Authors:**

Reviewer #1: The authors have comprehensively revised and strengthened the manuscript, with particular focus on the introduction and discussion. They have addressed my concerns and I consider the revised manuscript acceptable for publication.

Reviewer #2: Thank you for taking the time to address my comments. This paper has a coherent flow from start to finish and is ready for publication. One final thing, please state the computational time and machine employed to obtain the results for each model.

**Have the authors made all data and (if applicable) computational code underlying the findings in their manuscript fully available?**

Reviewer #1: Yes

Reviewer #2: Yes

PLOS authors have the option to publish the peer review history of their article (what does this mean?). If published, this will include your full peer review and any attached files.

Reviewer #1: No

Reviewer #2: No

---

## [Editor Report · Acceptance letter]

19 Apr 2024

PCOMPBIOL-D-23-01609R1 

Informing policy via dynamic models: Cholera in Haiti

Dear Dr Wheeler,

I am pleased to inform you that your manuscript has been formally accepted for publication in PLOS Computational Biology. Your manuscript is now with our production department and you will be notified of the publication date in due course.

With kind regards,

Judit Kozma
